# Transcoronary electrophysiological parameters in patients undergoing elective and acute coronary intervention

Rabeia Javid[1,2☯], Thomas A. Slater[1,2☯], Robert Bowes[1], Murugapathy Veerasamy[1], Nancy Wassef[1], Jennifer A. Rossington[1], Abdul M. Mozid[1], Ananth Kidambi[1], Stephen B. Wheatcroft[1,2], Muzahir H. Tayebjee[1,2]*

1 Department of Cardiology, Leeds Teaching Hospitals NHS Trust, Leeds, United Kingdom, 2 Leeds Institute of Cardiovascular and Metabolic Medicine, The University of Leeds, Leeds, United Kingdom

☯ These authors contributed equally to this work.
* muzahir.tayebjee@nhs.net

## Abstract

**Data Availability Statement:** All relevant data are within the paper and its Supporting Information files.

### Introduction

Percutaneous coronary intervention is performed routinely in the management of myocardial infarction with obstructive coronary disease, but intervention to arteries supplying nonviable myocardium may be harmful. It is important therefore to establish myocardial viability, and there is an unmet need in current clinical practice for real time viability assessment to aid in decision making. Transcoronary pacing to assess myocardial electrophysiological parameters may be a novel viability assessment technique which could be used in this regard.

### Methods

Coronary intervention was carried out according to standard departmental procedure with standard equipment. An exchange length coronary guidewire was passed into both target and reference coronary vessels and an over-the-wire balloon or microcatheter was used to insulate the guidewire and allow electrophysiological parameters to be assessed. Readings were obtained from all major epicardial vessels and substantial branches. At each position, an intracoronary electrocardiogram was recorded, and R wave amplitude was measured. Transcoronary pacing was then performed to establish threshold and impedance for each myocardial segment.

A viability cardiac MRI scan was performed for each patient. A standard segmental model was used to determine viability in each segment using an 'infarct score' based on degree of late gadolinium enhancement. Studies were reported blinded to the electrical parameters obtained from the coronary guidewire.

The primary outcome was the relationship between pacing threshold and myocardial segment infarct score. Secondary outcomes included the relationship between segmental infarct score and R wave height, and between segmental infarct score and pacing

**Funding:** RJ received fellowship funding from Abbott, https://www.abbott.co.uk/. The funders had no role in study design, data collection and analysis, decision to publish, or preparation of the manuscript.

**Competing interests:** RJ is an Abbott research fellow. MHT has received research grants from Abbott Medical, Biosense Webster and Medtronic. All other authors have declared that no competing interests exist. This does not alter our adherence to PLOS ONE policies on sharing data and materials.

impedance. Data were collected on the feasibility of studying the coronary segments as well as safety.

## Results

Sixty-five patients presenting with stable coronary artery disease or acute coronary syndromes to Leeds General Infirmary between September 2019 and August 2021 were included in the study. Electrophysiological parameters from segments with an infarct score of zero were obtained, with wide variances seen, with no significant difference in impedance or threshold in any territory. There was a significant difference in sensitivity for segments in the right coronary artery territory for both elective and acute patients. This likely relates to reduced myocardial mass in these territories. No significant association between infarct score and sensitivity, impedance or threshold were seen.

## Conclusion

This study has established intracoronary electrophysiological parameters in both normal myocardium and areas of myocardial scar. No reliable association was seen between impedance, threshold or R wave amplitude and degree of myocardial viability, contrasting with prior findings from our group and others. More work is therefore required to fully understand the role of transcoronary pacing in this setting.

## Introduction

Percutaneous coronary intervention (PCI) is performed routinely in the management of myocardial infarction with obstructive coronary disease. It is accepted, however, that coronary intervention to arteries supplying nonviable myocardium due to transmural scar does not confer benefit, and in some cases may be harmful [1,2].

Myocardial viability is currently assessed through a variety of non-invasive methods, including stress echocardiography, Fluorodeoxyglucose-positron emission tomography and cardiac magnetic resonance imaging (CMR) [3]. Although these imaging methods are safe and reliable, they risk introducing delay to revascularisation, and are infrequently accessible prior to angiography. Therefore, developing methods for viability assessment that could avoid delays to coronary intervention and offer real time assessment to aid in PCI decision making represents an unmet need in current clinical practice.

Transcoronary pacing (TCP) has been used effectively to treat bradyarrhythmias during coronary intervention in porcine models and small human trials [4–8]. Intracoronary electrocardiograms (IcECG) obtained from a guidewire tip have also been used to assess myocardial viability in an experimental setting [9–11]. Our group have published a feasibility study for the use of TCP to establish electrophysiological parameters of myocardium during coronary intervention and compared these parameters to CMR assessment of viability. This study found that myocardial impedance and pacing thresholds were significantly different between normal myocardium and myocardium with >50% mural scar on CMR, with no significant difference seen in R wave amplitude for any of the assessed groups [12]. As this was a pilot study, more data were required before definitive conclusions could be made on the effectiveness of this novel viability assessment technique.

This present study aims to expand upon the findings of our pilot study. It is the first to systematically examine transcoronary electrophysiological parameters in unselected patients and determine if these can be used to predict myocardial viability in the context of coronary intervention.

## Study aim

We hypothesised that TCP and IcECG analysis can be utilised to accurately predict myocardial viability and offer an 'on table' assessment of myocardial viability which could be utilised to guide coronary intervention. The study aims were to:

1. Determine the transcoronary electrophysiological parameters of myocardial segments in a population of patients undergoing coronary intervention

2. Determine if electrophysiological parameters can be used to predict myocardial viability compared to the current accepted standard of cardiac MRI

## Methods

A single-centre, prospective study was conducted at Leeds Teaching Hospitals NHS Trust. Ethical approval was obtained from the independent NHS Research Ethics Committee (REC) Wales REC 4 and the Leeds Teaching Hospitals Trust Research and Innovation Department. A total of sixty-five patients presenting with stable coronary artery disease or acute coronary syndromes to Leeds General Infirmary between September 2019 and August 2021 were included in the study. Each participant provided written informed consent and signed consent forms were stored securely. MRI studies and transcoronary pacing parameters were reported blinded to each other. Individual participants were not identifiable to authors responsible for data analysis during or after the data collection process.

Male and female patients between the ages of 18 and 99 were included. Patients required to have been listed for PCI, either electively with an established diagnosis of stable angina, or acutely following an admission for non-ST elevation myocardial infarction (NSTEMI). Exclusion criteria included patients unable to provide informed consent; patients deemed to be in the terminal stage of illness; pregnancy; haemodynamic instability; intervention for acute ST elevation MI; co-existing persistent atrial fibrillation with uncontrolled ventricular response; prior coronary artery bypass grafting; contraindications to PCI; claustrophobia; the presence of a permanent pacemaker and co-existent therapy with a class 1 or class 3 antiarrhythmic agent.

## Coronary intervention and transcoronary pacing

Coronary intervention was carried out according to standard departmental procedure with standard equipment. Route of access, pharmacological agents and type of stent used was at the discretion of the operator in accordance with LTHT departmental policy and published guidelines [13].

Transcoronary pacing required the following variations to standard coronary angiography.

During ECG skin electrode placement, a grounding patch was applied to the patient's back overlying the lumbar spine with the caudal part of the patch at the level of the posterior iliac spinous process.

An exchange length coronary guidewire (Asahi Sion Blue) was passed into both the target and reference coronary vessels to allow the use of either an over-the-wire balloon (Boston

Scientific, US) or microcatheter, depending on procedure performed. The Sion Blue guidewire was used in each case to ensure consistency, as previous studies have shown different wires have different conduction properties [5], and our previous pilot study demonstrated adequate conduction with the Sion Blue [12], which is the most commonly used guidewire in our institution. Use of an over-the-wire balloon or microcatheter ensured electrical insulation of the guidewire, which cannot be achieved using monorail balloons. The distal tip of the guidewire was exposed beyond the balloon/microcatheter by one centimetre, to achieve adequate electrical contact, as used in our pilot study [12]. The proximal end of the guidewire was placed in its holder to maintain electrical insulation. A small segment of wire was exposed and clipped to the pacing programmer (Abbott Medical, US). A second clip was applied to the grounding patch, forming a unipolar pacing circuit.

The guidewire was advanced to the distal part of each coronary artery for initial recordings. The distal tip was then advanced sequentially into accessible proximal side branches selected by the operator. Fluoroscopy was used to confirm the location of the wire-balloon unit and readings were obtained from all accessible major epicardial vessels and substantial branches, corresponding to AHA defined myocardial segments [14]. At each position, an intra coronary electrocardiogram (icECG) was recorded on paper and R wave amplitude was measured from this. Transcoronary pacing was then performed to establish pacing threshold and impedance for each myocardial segment.

## Cardiac MRI

A viability cardiac MRI scan was performed for each patient. Imaging was performed at any time point and could be carried out before or after coronary intervention, with operators blinded to cardiac MRI findings if performed prior to coronary intervention. Minimum dataset requirements included cardiac anatomy, resting left ventricular function and late gadolinium enhancement, and a standard segmental model was used to determine viability in each segment using an 'infarct score' based on degree of late gadolinium enhancement. A score of 'zero' indicated fully viable myocardium with no scar; 'one' indicated 1–25% subendothelial scar; 'two' indicated 26–50% scar; 'three' indicated 51–75% scar and 'four' 76–100% scar. Scans were carried out using 1.5 or 3 Tesla scanners with dedicated cardiac coils at Leeds General Infirmary.

## Outcomes

The primary outcome was the relationship between pacing threshold and myocardial segment infarct score. Secondary outcomes included the relationship between segmental infarct score and R wave height, and between segmental infarct score and pacing impedance. Data were collected on the feasibility of studying the coronary segments as well as safety.

## Statistical analysis

Data are expressed as means ± standard deviation or median (IQR). Electrophysiological sensitivity, impedance, and threshold data were compared with scar burden seen on cardiac MRI using logged ANOVA.

Power analyses were completed using G*Power (version 3.0.10). These assumed an equal number of segments in each group (<50% scar and >50% scar) and permitted a probable error rate (α) of 0.05. Glass's delta was calculated for pacing threshold using the mean difference between <50% scar and >50% scar for each measure (1.06V;(12)), to ensure power to detect the smallest effect size. Detection of a 0.38V difference in pacing threshold with 80% power required 220 segment measurements.

## Results and discussion

A total of 65 patients were recruited into the trial, 40 for elective PCI and 25 acutely in the context of NSTEMI. Baseline parameters for trial participants, including relevant co-morbidities, current medication and angiographic findings are shown in Table 1. No procedural complications were encountered relating to trans-coronary pacing.

Segmental analysis by TCP was possible in all patients recruited, with 369 total segments analysed, and 36 segments excluded from final analysis. Exclusion occurred after initial analysis and following review of acquired angiographic fluoroscopic images. Twenty-five segments were excluded due to inadequate guidewire engagement of the target vessel, and the remaining eleven segments were excluded due to inadequate insulation of the distal guidewire with either an over the wire balloon or microcatheter, as discussed previously in the methods section. It was not possible to analyse every myocardial segment in each patient, due to differences in coronary anatomy limiting accessibility, with a mean of six segments per patient analysed.

The majority of analysed segments were given an infarct score of 0 on MRI assessment, indicating fully viable territory with no evidence of scar. These normal segments were then grouped based on epicardial coronary artery territory, either left anterior descending (LAD), circumflex (Cx) or right coronary artery (RCA). The electrophysiological parameters obtained are outlined below for elective and acute patients, in Table 2A and 2B respectively. Widely

**Table 1. Baseline parameters and angiography details for elective and acute trial participants.**

| | Elective | Acute |
|---|---|---|
| | (n = 40) | (n = 25) |
| Age (years +/- 1 SD) | 61 +/- 8 | 63 +/- 12 |
| Male n (%) | 34 (85) | 21 (85) |
| Hypertension n (%) | 19 (47) | 2 (8) |
| Diabetes n (%) | 13 (32) | 5 (20) |
| Previous MI n (%) | 33 (82) | 10 (40) |
| Previous PCI n (%) | 26 (65) | 8 (32) |
| Creatinine (mmol/l +/- 1 SD) | 85 (25) | 76 (17) |
| LV function (EF >45%:<45%) | 26:14 | 18:7 |
| Atrial fibrillation n (%) | 0 (0) | 3 (12) |
| Beta blocker n (%) | 34 (85) | 25 (100) |
| Calcium channel antagonist n (%) | 13 (32) | 5 (20) |
| ACE inhibitor n (%) | 32 (80) | 24 (96) |
| Statin n (%) | 38 (95) | 23 (92) |
| Troponin (IU/l +/- 1 SD) | | 6042 +/- 9379 |
| Findings on angiography | | |
| Left main disease n (%) | 3 (7) | 4 (16) |
| Single vessel n (%) | 19 (47) | 7 (28) |
| Double vessel n (%) | 13 (32) | 16 (64) |
| Triple vessel n (%) | 1 (0.02) | 2 (8) |
| PCI performed n (%) | 31 (77) | 21 (84) |
| Total procedure duration (minutes +/- 1 SD) | 106 +/- 47 | 80 +/- 27 |
| Total TCP time (minutes +/- 1 SD) | 20 +/- 9 | 24 +/- 9 |
| Total fluoroscopy time (minutes +/- 1 SD) | 30 +/- 17 | 22 +/- 10 |
| Total fluoroscopy dose dose (cGycm2 +/- 1 SD) | 8595 +/- 4954 | 7375 +/- 4308 |
| Fluoroscopy time for TCP (minutes +/- 1 SD) | 8 +/- 4 | 8 +/- 5 |
| Fluoroscopy dose for TCP (cGycm2 +/- 1 SD) | 1850 +/- 1586 | 2000 +/- 1793 |

**Table 2. A. Electrophysiological parameters of normal segments in elective patients with stable angina.** A significant reduction in sensitivity was seen in the right coronary artery (RCA) territory compared to the left anterior descending (LAD) and circumflex (Cx) territories (p = 0.001). **B.** Electrophysiological parameters of normal segments in acute patients hospitalised with NSTEMI. A significant reduction in sensitivity was seen in the RCA territory compared to the LAD and Cx territories (p = <0.001).

|  | Art territory | n | Q1 | Median | Q3 | p |
|---|---|---|---|---|---|---|
| Sensitivity | LAD | 72 | 4.8 | 7.4 | 11.575 | |
| (mV) | Cx | 48 | 4.025 | 8.05 | 12.4 | |
|  | RCA | 34 | 3.225 | 4.35* | 6.675 | 0.001 |
| Impedance | LAD | 72 | 320.8 | 380 | 446.8 | |
| (Ohms) | Cx | 48 | 312 | 359 | 533 | |
|  | RCA | 34 | 314 | 368 | 483 | 0.186 |
| Threshold | LAD | 72 | 1.925 | 3.15 | 6 | |
| (V) | Cx | 48 | 2.175 | 4.2 | 6.375 | |
|  | RCA | 34 | 1.975 | 3.85 | 8 | 0.667 |
|  | Art Territory | n | Q1 | Median | Q3 | p |
| Sensitivity | LAD | 58 | 6 | 9.05 | 13.425 | |
| (mV) | Cx | 41 | 4.15 | 7.6 | 11.75 | |
|  | RCA | 30 | 2.8 | 4.85* | 6.2 | <0.001 |
| Impedance | LAD | 58 | 324 | 419 | 523 | |
| (Ohms) | Cx | 41 | 337 | 422 | 676 | |
|  | RCA | 30 | 292 | 389 | 574 | 0.535 |
| Threshold | LAD | 58 | 1.8 | 3 | 5.125 | |
| (V) | Cx | 41 | 1.65 | 3.1 | 5.75 | |
|  | RCA | 30 | 2.18 | 3.5 | 5.75 | 0.258 |

ranging values were seen, with no significant difference in impedance or threshold seen in any territory. The only parameters which demonstrated a significant difference were sensitivity values obtained from segments in the RCA territory for both elective and acute patients. This likely relates to reduced myocardial mass in the right ventricle, supplied by the RCA.

A comparison of electrophysiological parameters depending on infarct score as assessed by CMR are shown in Table 3A and 3B for elective and acute patients respectively. No significant association between infarct score and sensitivity, impedance or threshold were seen.

This study is the first to systematically examine transcoronary electrophysiological parameters in unselected patients and determine if these can be used to predict myocardial viability in the context of coronary intervention, compared to the currently accepted standard of cardiac MRI. This current study builds on the findings from our prior feasibility study which found that myocardial impedance and pacing threshold could be used to differentiate between normal and scarred myocardium in the LAD territory, although as a pilot study this was not powered to draw any definitive conclusions [12].

In contrast to our prior findings, when a larger unselected population was examined and data from all three coronary territories was assessed, there was no relationship found between impedance or threshold measurements and degree of myocardial scarring. The only finding which reached statistical significance was an association between reduced R wave amplitude in the RCA territory in normal segments, likely related to reduced myocardial mass in the right ventricle.

It is noteworthy that, apart from the R wave amplitude changes in RCA territory noted above, there was no significant difference seen in electrophysiological parameters between normal myocardium with an infarct score of zero and myocardium that was judged on cardiac MRI to be entirely non-viable, with an infarct score of four. These data do not align with our

**Table 3. A. CMR infarct score and respective electrophysiological parameters in segments assessed during elective coronary angiography for stable angina. B.** CMR infarct score and respective electrophysiological parameters in segments assessed during acute coronary angiography following NSTEMI.

| **Elective LAD** | Infarct Score | n | Q1 | Median | Q3 | p |
|---|---|---|---|---|---|---|
| Sensitivity | 0 | 72 | 4.8 | 7.4 | 11.575 | |
| (mV) | 1 | 7 | 5.2 | 5.8 | 17.3 | |
| | 2 | 3 | 4.6 | 5.1 | 7.8 | |
| | 3 | 4 | 1.38 | 6.65 | 13.35 | |
| | 4 | 3 | 4.9 | 6.3 | 8.3 | 0.172 |
| Impedance | 0 | 72 | 320.8 | 380 | 446.8 | |
| (Ohms) | 1 | 7 | 347 | 382 | 418 | |
| | 2 | 3 | 279 | 351 | 404 | |
| | 3 | 4 | 238 | 333.5 | 413.3 | |
| | 4 | 3 | 310 | 337 | 379 | 0.766 |
| Threshold | 0 | 72 | 1.925 | 3.15 | 6 | |
| (V) | 1 | 7 | 2 | 4 | 7.5 | |
| | 2 | 3 | 1.6 | 2.1 | 6 | |
| | 3 | 4 | 1.53 | 4.35 | 6.88 | |
| | 4 | 3 | 1.5 | 3.3 | 4.5 | 0.947 |
| **Elective Cx** | Infarct Score | n | Q1 | Median | Q3 | p |
| Sensitivity | 0 | 48 | 4.025 | 8.05 | 12.4 | |
| (mV) | 1 | 9 | 2.8 | 4.3 | 9.1 | |
| | 2 | 2 | * | 9.6 | * | |
| | 3 | 1 | * | 2.1 | * | 0.071 |
| Impedance | 0 | 48 | 312 | 359 | 533 | |
| (Ohms) | 1 | 9 | 283 | 365 | 2735 | |
| | 2 | 2 | * | 276.5 | * | |
| | 3 | 1 | * | 286 | * | 0.522 |
| Threshold | 0 | 48 | 2.175 | 4.2 | 6.375 | |
| (V) | 1 | 9 | 3.15 | 10 | 15 | |
| | 2 | 2 | * | 5.5 | * | |
| | 3 | 1 | * | 1.2 | * | 0.134 |
| **Elective RCA** | Infarct Score | n | Q1 | Median | Q3 | p |
| Sensitivity | 0 | 34 | 3.225 | 4.35 | 6.675 | |
| (mV) | 1 | 13 | 2.45 | 3.9 | 5.8 | |
| | 2 | 7 | 2.8 | 4.1 | 4.7 | |
| | 3 | 3 | 3.7 | 6.5 | 6.9 | |
| | 4 | 6 | 3.13 | 4.1 | 7 | 0.573 |
| Impedance | 0 | 34 | 314 | 368 | 483 | |
| (Ohms) | 1 | 13 | 268.5 | 328 | 420 | |
| | 2 | 7 | 293 | 327 | 470 | |
| | 3 | 3 | 264 | 360 | 404 | |
| | 4 | 6 | 329.8 | 401 | 554 | 0.33 |
| Threshold | 0 | 34 | 1.975 | 3.85 | 8 | |
| (V) | 1 | 13 | 4.75 | 6 | 8.5 | |
| | 2 | 7 | 6.5 | 9 | 20 | |
| | 3 | 3 | 3 | 3.4 | 20 | |
| | 4 | 6 | 0.98 | 4.15 | 7.88 | 0.102 |
| **Acute LAD** | Infarct Score | n | Q1 | Median | Q3 | p |
| Sensitivity | 0 | 58 | 6 | 9.05 | 13.425 | |

(*Continued*)

**Table 3.** (Continued)

| | Infarct Score | n | Q1 | Median | Q3 | p |
|---|---|---|---|---|---|---|
| (mV) | 1 | 7 | 5.6 | 7.1 | 11.7 | |
| | 2 | 1 | * | 10.9 | * | |
| | 3 | 1 | * | 4 | * | |
| | 4 | 2 | * | 7.8 | * | 0.711 |
| Impedance | 0 | 58 | 324 | 419 | 523 | |
| (Ohms) | 1 | 7 | 254 | 339 | 487 | |
| | 2 | 1 | * | 345 | * | |
| | 3 | 1 | * | 515 | * | |
| | 4 | 2 | * | 327 | * | 0.694 |
| Threshold | 0 | 58 | 1.8 | 3 | 5.125 | |
| (V) | 1 | 7 | 0.9 | 1.8 | 4.5 | |
| | 2 | 1 | * | 1.2 | * | |
| | 3 | 1 | * | 5 | * | |
| | 4 | 2 | * | 3.3 | * | 0.4 |
| **Acute Cx** | Infarct Score | n | Q1 | Median | Q3 | p |
| Sensitivity | 0 | 41 | 4.15 | 7.6 | 11.75 | |
| (mV) | 1 | 2 | * | 8.2 | * | |
| | 3 | 4 | 2.775 | 3.3 | 4.575 | 0.202 |
| Impedance | 0 | 41 | 337 | 422 | 676 | |
| (Ohms) | 1 | 2 | * | 303 | * | |
| | 3 | 4 | 238 | 255.5 | 346.5 | 0.17 |
| Threshold | 0 | 41 | 1.65 | 3.1 | 5.75 | |
| (V) | 1 | 2 | * | 2.75 | * | |
| | 3 | 4 | 2.55 | 4.15 | 8.75 | 0.61 |
| **Acute RCA** | Infarct Score | n | Q1 | Median | Q3 | p |
| Sensitivity | 0 | 30 | 2.8 | 4.85 | 6.2 | |
| (mV) | 1 | 2 | * | 12 | * | |
| | 2 | 5 | 3.9 | 4.6 | 9.25 | |
| | 3 | 4 | 2.75 | 3.35 | 4.325 | 0.213 |
| Impedance | 0 | 30 | 292 | 389 | 574 | |
| (Ohms) | 1 | 2 | * | 524 | * | |
| | 2 | 5 | 216 | 282 | 2174 | |
| | 3 | 4 | 284 | 304 | 448.5 | 0.768 |
| Threshold | 0 | 30 | 2.18 | 3.5 | 5.75 | |
| (V) | 1 | 2 | * | 6.5 | * | |
| | 2 | 5 | 1.9 | 3 | 3.3 | |
| | 3 | 4 | 1.8 | 3.4 | 5.075 | 0.394 |

feasibility study, or prior studies which have noted significant differences in IcECG parameters between viable and non-viable myocardium [10,11]. There is a possibility therefore that individual operator technique impacted on these findings, or that further work is required to identify more reliable hardware. Additionally, the wide variance in electrophysiological parameters obtained from normal segments meant that any potential differences seen in infarcted myocardium were less likely to be of significance, reducing the likelihood of TCP being an effective tool in the assessment of viability. It is unclear why such wide variances in readings were seen, but they may relate to inconsistency in contact, pacing areas of epicardial fat rather than myocardium, or inter-operator variability. Additionally, coronary anatomy variance meant it was

not possible to analyse every myocardial segment in each patient, limiting the use of TCP in the analysis of myocardial viability in instances such as chronic total occlusion of an epicardial coronary artery, a situation where myocardial viability is an important factor in the decision whether to attempt complex revascularisation.

Finally, it is of interest that it was possible to obtain myocardial capture reliably even in segments with transmural scar, indicating the presence of viable myocardium within transmural scar. This raises the possibility that TCP may be of use in predicting the risk of myocardial re-entry mediated ventricular tachycardia in areas of transmural scar and may add to current methods for identifying potential sites of the critical isthmus in such cases [15].

## Conclusion

This study has established intracoronary electrophysiological parameters in both normal myocardium and areas of myocardial scar. A difference in sensitivity in normal segments in the RCA territory was seen, likely related to reduced corresponding myocardial mass compared to other coronary territories. No reliable association was seen between impedance, threshold or R wave amplitude and degree of myocardial viability, contrasting with prior findings from our group and others.

The inability to differentiate normal myocardium from areas of scar may be related to several issues, including low segment numbers for higher infarct scores; wide distribution of values in normal segments and a lack of consistency in establishing contact. More work is therefore required to fully understand the role of transcoronary pacing in this setting, but our study findings have raised the possibility of alternative uses, such as in the prediction and management of scar related ventricular tachycardia.

## Supporting information

**S1 Raw data.**
(XLSX)

## Author Contributions

**Conceptualization:** Muzahir H. Tayebjee.

**Data curation:** Rabeia Javid, Robert Bowes, Murugapathy Veerasamy, Nancy Wassef, Abdul M. Mozid, Ananth Kidambi.

**Formal analysis:** Rabeia Javid.

**Funding acquisition:** Muzahir H. Tayebjee.

**Investigation:** Rabeia Javid, Murugapathy Veerasamy, Nancy Wassef, Jennifer A. Rossington, Abdul M. Mozid, Ananth Kidambi, Stephen B. Wheatcroft, Muzahir H. Tayebjee.

**Methodology:** Muzahir H. Tayebjee.

**Project administration:** Robert Bowes, Murugapathy Veerasamy, Nancy Wassef.

**Resources:** Muzahir H. Tayebjee.

**Supervision:** Muzahir H. Tayebjee.

**Writing – original draft:** Rabeia Javid, Thomas A. Slater.

**Writing – review & editing:** Rabeia Javid, Thomas A. Slater, Jennifer A. Rossington, Stephen B. Wheatcroft, Muzahir H. Tayebjee.

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
