## [Decision Letter · Decision Letter 0]

7 Dec 2022

PONE-D-22-28759Transcoronary electrophysiological parameters in patients undergoing elective and acute coronary interventionPLOS ONE

Dear Dr. Tayebjee,

Thank you for submitting your manuscript to PLOS ONE. After careful consideration, we feel that it has merit but does not fully meet PLOS ONE’s publication criteria as it currently stands. Therefore, we invite you to submit a revised version of the manuscript that addresses the points raised during the review process.

Please address each comment from both reviewers in an individual fashion in the response letter.

We look forward to receiving your revised manuscript.

Kind regards,

R. Jay Widmer

Academic Editor

PLOS ONE

Journal Requirements:

When submitting your revision, we need you to address these additional requirements. 1. Please ensure that your manuscript meets PLOS ONE's style requirements, including those for file naming. The PLOS ONE style templates can be found at https://journals.plos.org/plosone/s/file?id=wjVg/PLOSOne_formatting_sample_main_body.pdf and https://journals.plos.org/plosone/s/file?id=ba62/PLOSOne_formatting_sample_title_authors_affiliations.pdf 2. Thank you for sending the ethics approval document as requested. We note that the document refers to an amendment. We therefore ask you that you please specify the date when you received your first ethics approval. Thank you for your attention to this request. 3. Thank you for stating the following in the Competing Interests section: "RJ is an Abbott research fellow. MHT has received research grants from Abbott Medical, Biosense Webster and Medtronic. All other authors have declared that no competing interests exist." Please confirm that this does not alter your adherence to all PLOS ONE policies on sharing data and materials, by including the following statement: ""This does not alter our adherence to  PLOS ONE policies on sharing data and materials.” (as detailed online in our guide for authors http://journals.plos.org/plosone/s/competing-interests).  If there are restrictions on sharing of data and/or materials, please state these. Please note that we cannot proceed with consideration of your article until this information has been declared.  Please include your updated Competing Interests statement in your cover letter; we will change the online submission form on your behalf. 4. In your Data Availability statement, you have not specified where the minimal data set underlying the results described in your manuscript can be found. PLOS defines a study's minimal data set as the underlying data used to reach the conclusions drawn in the manuscript and any additional data required to replicate the reported study findings in their entirety. All PLOS journals require that the minimal data set be made fully available. For more information about our data policy, please see http://journals.plos.org/plosone/s/data-availability. Upon re-submitting your revised manuscript, please upload your study’s minimal underlying data set as either Supporting Information files or to a stable, public repository and include the relevant URLs, DOIs, or accession numbers within your revised cover letter. For a list of acceptable repositories, please see http://journals.plos.org/plosone/s/data-availability#loc-recommended-repositories. Any potentially identifying patient information must be fully anonymized. Important: If there are ethical or legal restrictions to sharing your data publicly, please explain these restrictions in detail. Please see our guidelines for more information on what we consider unacceptable restrictions to publicly sharing data: http://journals.plos.org/plosone/s/data-availability#loc-unacceptable-data-access-restrictions. Note that it is not acceptable for the authors to be the sole named individuals responsible for ensuring data access. We will update your Data Availability statement to reflect the information you provide in your cover letter. 5. We note that you have stated that you will provide repository information for your data at acceptance. Should your manuscript be accepted for publication, we will hold it until you provide the relevant accession numbers or DOIs necessary to access your data. If you wish to make changes to your Data Availability statement, please describe these changes in your cover letter and we will update your Data Availability statement to reflect the information you provide. 6. Please note that in order to use the direct billing option the corresponding author must be affiliated with the chosen institute. Please either amend your manuscript to change the affiliation or corresponding author, or email us at plosone@plos.org with a request to remove this option.****

Additional Editor Comments:

While mostly favorable, there are some concerns by the reviewers regarding the background information, methodology, implications, and overall quality of writing. We would suggest the authors carefully consider the individual comments from both reviewers, and address each comment in the corresponding cover letter.

Reviewers' comments:

Reviewer's Responses to Questions

**Comments to the Author**

1. Is the manuscript technically sound, and do the data support the conclusions?

Reviewer #1: No

Reviewer #2: Partly

2. Has the statistical analysis been performed appropriately and rigorously? 

Reviewer #1: No

Reviewer #2: Yes

3. Have the authors made all data underlying the findings in their manuscript fully available?

Reviewer #1: No

Reviewer #2: Yes

4. Is the manuscript presented in an intelligible fashion and written in standard English?

Reviewer #1: No

Reviewer #2: Yes

5. Review Comments to the Author

Reviewer #1: For authors;

This is a prospective study of patients with acute and chronic coronary disease submitted to angiographic study for detection of electrophysiological parameters in myocardial regions admitted as healthy or diseased. Authors found similar results in both groups.

Major problems:

1- Study is unethical because it is an “in anima nobile” experience

2-I have doubts that the authors would carry out this study on their own family members.

3-Study has several shortcomings: A: rational, there is no minimally scientific foundation on the application of the model, as well as the knowledge arising from it.B: Aiming to know the result of the stimulation and applying it in the viability of the myocyte is inadequate. C: Including acute and chronic patients in the same study is minimally ignoring the physiology and pathophysiology of the cardiac muscle.

4-absence of description of the investigation models both of the electrophysiological study and of the cardiac magnetic resonance.

5-The study is poorly written and its reading is a challenge.

Reviewer #2: The authors designed a study as a follow up to a prior internal pilot project, which had suggested that basic electrophysiologic measurements obtained from within the coronary artery by way of the coronary guidewire may correlate with myocardial viability at that segment.

The design of the trial is fundamentally simple and straightforward. A strength is the fact that all patients underwent MRI to provide high level data on the presence or absence of infarcted tissue in the gross distribution being studied. A weakness is that just because there is an infarcted inferior wall segment, that may or may not be the area precisely studied with pacing parameters.

The study was able to provide some baseline measurements for pacing threshold, impedance, or sensitivity. However, there was no relationship seen between the presence or absence of infarct and any of the three measured elements: pacing threshold, impedance, or R wave, thus the hypothesis that these simple derived electrophysiologic parameters would predict viability is not supported.

The paper is well written overall, with the exception of the tables which I think warrant some revisions to make them more quickly accessible to readers. (detailed below)

Suggestions:

1. Transcoronary pacing long predates cited articles 4-6. Consider adding reference to seminal articles including Meier et al PMID: 3156008 and Mixon et al PMID PMID: 15065145, the latter of which even includes some benchtop data on standard impedance and variations in wire effect.

2. Table 1 is confusing, and needs some additional explanation. E.g. Age (years) says 61(8). It is unclear what these numbers express: mean with a deviation? Similar comment applies to creatinine, PCI performed, troponin, and the last 6 entries (i.e. what do the two numbers represent?)

3. Table 1, the expression of LV function is clunky and ill defined. Perhaps a binary measurement would be easier to express and read, e.g. LV EF < 45%, yes or no

4. Table 2 and 2b need explanation for why one number is bolded and asterisked.

5. Please clarify more about how many coronary sections were studied in each patient. The methods section suggest that every AHA defined myocardial segment was studied, yet in fact back of the napkin calculations suggest there were perhaps 5 total measurement per patient on average. One wonders whether the limited number of sampling sites could affect the outcomes. If an MRI has a scar in a particular distribution (e.g. the inferolateral wall), and only one reading is taken from the circumflex, it could be that the area sampled does not correlate well with the infarct area.

6. Perhaps some discussion of wire composition and the impact that this could have on measurements. Again, as referenced in the Mixon article, various wires conduct very differently. Perhaps an explanation for why the Sion blue is felt to be an optimal wire for this type of work. The implication from the paper is that every study was done with the same manufactured wire, but this should be stated clearly if that is true. If not true, it introduces another source or variation in the derived measurements.

6. PLOS authors have the option to publish the peer review history of their article (what does this mean?). If published, this will include your full peer review and any attached files.

Reviewer #1: No

Reviewer #2: No

---

## [Author Response · Author response to Decision Letter 0]

5 Jan 2023

Thank you very much to the academic editor and both reviewers for taking the time to read and appraise our manuscript. We have outlined our responses below and in the separately attached 'response to reviewers' document. Please note in addition to the comments made, we noticed an error within the text for total number of segments analysed, which has now been amended and can be reviewed in our 'tracked changes' manuscript. This had no effect on the outcome of the study, and the tables have not required any alteration due to this amendment.

Kind Regards,

Dr Muzahir Tayebjee

Reviewer #1:

This is a prospective study of patients with acute and chronic coronary disease submitted to angiographic study for detection of electrophysiological parameters in myocardial regions admitted as healthy or diseased. Authors found similar results in both groups.

Major problems:

1- Study is unethical because it is an “in anima nobile” experience

Thank you for taking the time to review our paper. We received ethical approval prior to commencement of the study. Additionally, transcoronary pacing has been previously proven to be safe in both animal models and studies with human patients. We would therefore respectfully disagree, and we believe the study to be ethical.

2-I have doubts that the authors would carry out this study on their own family members.

This is an unusual comment to receive regarding a paper in which ethical approval has been received prior to commencement, and all participants were adults providing written informed consent. It seems too personal to require direct reply, however no concerns were received from any of the authors at any point during or after the study, and no complications occurred as a result of performing transcoronary pacing.

3-Study has several shortcomings: A: rational, there is no minimally scientific foundation on the application of the model, as well as the knowledge arising from it.B: Aiming to know the result of the stimulation and applying it in the viability of the myocyte is inadequate. C: Including acute and chronic patients in the same study is minimally ignoring the physiology and pathophysiology of the cardiac muscle.

4-absence of description of the investigation models both of the electrophysiological study and of the cardiac magnetic resonance.

5-The study is poorly written and its reading is a challenge.

We thank the reviewer again for taking the time to read our study and provide comments. 

A: We believe our rationale for the study to be adequately described in the introduction as shown below.

‘Although these imaging methods are safe and reliable, they risk introducing delay to revascularisation, and are infrequently accessible prior to angiography. Therefore, developing methods for viability assessment that could avoid delays to coronary intervention and offer real time assessment to aid in PCI decision making represents an unmet need in current clinical practice.

Transcoronary pacing (TCP) has been used effectively to treat bradyarrhythmias during coronary intervention in porcine models and small human trials.(4)(5)(6) Intracoronary electrocardiograms (IcECG) obtained from a guidewire tip have also been used to assess myocardial viability in an experimental setting.(7)(8)(9) Our group have published a feasibility study for the use of TCP to establish electrophysiological parameters of myocardium during coronary intervention and compared these parameters to CMR assessment of viability.’

There have been several prior studies examining transcoronary pacing, and our own pilot study proved its feasibility. It is established that viable myocardium will have different physiological parameters to non-viable myocardium, and we therefore believe the scientific foundation to be present, with our present study building on our understanding of the physiological parameters of viable and non-viable myocardium.

B: Unfortunately we do not fully understand this comment. We would be happy to address it if it could be clarified.

C: We completely agree that acute and chronic coronary syndromes involve different intracoronary pathophysiological processes. However, assessment of myocardial viability prior to revascularisation is relevant to both and so we believe it was appropriate to include patients presenting with both syndromes in our study. As each group were analysed and reported separately, we do not feel including both groups impacted on our findings.

4: Unfortunately we do not fully understand this comment. We would be happy to address it if it could be clarified.

5: We are sorry the reviewer found reading our study to be a challenge. We welcome the second reviewer’s comments that our study was well written, and the comments from the editorial team that our study has merit.

Reviewer #2: 

The authors designed a study as a follow up to a prior internal pilot project, which had suggested that basic electrophysiologic measurements obtained from within the coronary artery by way of the coronary guidewire may correlate with myocardial viability at that segment.

The design of the trial is fundamentally simple and straightforward. A strength is the fact that all patients underwent MRI to provide high level data on the presence or absence of infarcted tissue in the gross distribution being studied. A weakness is that just because there is an infarcted inferior wall segment, that may or may not be the area precisely studied with pacing parameters.

The study was able to provide some baseline measurements for pacing threshold, impedance, or sensitivity. However, there was no relationship seen between the presence or absence of infarct and any of the three measured elements: pacing threshold, impedance, or R wave, thus the hypothesis that these simple derived electrophysiologic parameters would predict viability is not supported.

The paper is well written overall, with the exception of the tables which I think warrant some revisions to make them more quickly accessible to readers. (detailed below)

Suggestions:

1. Transcoronary pacing long predates cited articles 4-6. Consider adding reference to seminal articles including Meier et al PMID: 3156008 and Mixon et al PMID PMID: 15065145, the latter of which even includes some benchtop data on standard impedance and variations in wire effect.

Thank you for recognising the strengths of our study, and for your valuable comments to help improve our paper. We agree these are important references and have now included them. 

2. Table 1 is confusing, and needs some additional explanation. E.g. Age (years) says 61(8). It is unclear what these numbers express: mean with a deviation? Similar comment applies to creatinine, PCI performed, troponin, and the last 6 entries (i.e. what do the two numbers represent?)

Thank you for highlighting that this table is not clear in what each number represents. The numbers are mean plus standard deviation, and the table has been amended to reflect this more clearly.

3. Table 1, the expression of LV function is clunky and ill defined. Perhaps a binary measurement would be easier to express and read, e.g. LV EF < 45%, yes or no

Thank you for your comment. We have now amended Table 1 as you suggest, with a binary measurement of LVEF either above or below 45%.

4. Table 2 and 2b need explanation for why one number is bolded and asterisked.

Thank you for highlighting that this is not clear. The asterisked number in bold represents the significance of this result. We have now removed the bold highlighting and explained within the figure legends that a significant difference in RCA sensitivity was seen. 

5. Please clarify more about how many coronary sections were studied in each patient. The methods section suggest that every AHA defined myocardial segment was studied, yet in fact back of the napkin calculations suggest there were perhaps 5 total measurement per patient on average. One wonders whether the limited number of sampling sites could affect the outcomes. If an MRI has a scar in a particular distribution (e.g. the inferolateral wall), and only one reading is taken from the circumflex, it could be that the area sampled does not correlate well with the infarct area.

Thank you for noting this, and on reflection we agree it is not clear from the text how many sections were analysed in each patient. It was not possible to analyse all segments due to differences in coronary anatomy, and this represents a limitation of our study. We have amended the text accordingly as shown below.

‘It was not possible to analyse every myocardial segment in each patient, due to differences in coronary anatomy limiting accessibility, with a mean of six segments per patient analysed.’

‘Additionally, coronary anatomy variance meant it was not possible to analyse every myocardial segment in each patient, limiting the use of TCP in the analysis of myocardial viability in instances such as chronic total occlusion of an epicardial coronary artery, a situation where myocardial viability is an important factor in the decision whether to attempt complex revascularisation.’

6. Perhaps some discussion of wire composition and the impact that this could have on measurements. Again, as referenced in the Mixon article, various wires conduct very differently. Perhaps an explanation for why the Sion blue is felt to be an optimal wire for this type of work. The implication from the paper is that every study was done with the same manufactured wire, but this should be stated clearly if that is true. If not true, it introduces another source or variation in the derived measurements.

Thank you for noting this. Every study was indeed performed with the Sion Blue wire, which is the guidewire most commonly used in our institution, and so most familiar to the operators. Our previous pilot study had analysed different types of wire and found the Sion Blue to adequately conduct. We therefore used this wire exclusively to remove the possibility of hardware related variability, and to avoid introducing unfamiliar equipment. We have amended the text within the Methods section accordingly, as shown below.

‘The Sion Blue guidewire was used in each case to ensure consistency, as previous studies have shown different wires have different conduction properties,(5) and our previous pilot study demonstrated adequate conduction with the Sion Blue,(12) which is the most commonly used guidewire in our institution.’

---

## [Decision Letter · Decision Letter 1]

23 Jan 2023

Transcoronary electrophysiological parameters in patients undergoing elective and acute coronary intervention

PONE-D-22-28759R1

Dear Dr. Tayebjee,

We’re pleased to inform you that your manuscript has been judged scientifically suitable for publication and will be formally accepted for publication once it meets all outstanding technical requirements.

Kind regards,

R. Jay Widmer

Academic Editor

PLOS ONE

Additional Editor Comments (optional):

The authors have adequately addressed all pertinent comments from the reviewers.

Reviewers' comments:

Reviewer's Responses to Questions

**Comments to the Author**

1. If the authors have adequately addressed your comments raised in a previous round of review and you feel that this manuscript is now acceptable for publication, you may indicate that here to bypass the “Comments to the Author” section, enter your conflict of interest statement in the “Confidential to Editor” section, and submit your "Accept" recommendation.

Reviewer #2: All comments have been addressed

2. Is the manuscript technically sound, and do the data support the conclusions?

Reviewer #2: Yes

3. Has the statistical analysis been performed appropriately and rigorously? 

Reviewer #2: Yes

4. Have the authors made all data underlying the findings in their manuscript fully available?

Reviewer #2: Yes

5. Is the manuscript presented in an intelligible fashion and written in standard English?

Reviewer #2: Yes

6. Review Comments to the Author

Reviewer #2: Thank you for your revisions. I am satisified that all of my concerns have been adequately answered.

7. PLOS authors have the option to publish the peer review history of their article (what does this mean?). If published, this will include your full peer review and any attached files.

Reviewer #2: No

---

## [Editor Report · Acceptance letter]

26 Jan 2023

PONE-D-22-28759R1 

Transcoronary electrophysiological parameters in patients undergoing elective and acute coronary intervention 

Dear Dr. Tayebjee:

I'm pleased to inform you that your manuscript has been deemed suitable for publication in PLOS ONE. Congratulations! Your manuscript is now with our production department. 

Kind regards, 

on behalf of

Dr. R. Jay Widmer 

Academic Editor

PLOS ONE